# Non-pharmacological labor pain management practice and associated factors among skilled attendants working in public health facilities in Gamo and Gofa zone, Southern Ethiopia: A cross-sectional study

**Biresaw Wassihun**[ID][1]*, **Yosef Alemayehu**[ID][1], **Teklemariam Gultie**[1], **Beemnet Tekabe**[2], **Birhaneselasie Gebeyehu**[3☯]

**1** Department of Midwifery, College of Medicine and Health Science, Arba Minch University, Arba Minch, Ethiopia, **2** Department of Health Informatics, College of Medicine and Health Science, Arba Minch University, Arba Minch, Ethiopia, **3** Department of Medical Nursing, School of Nursing, College of Medicine and Health Science, University of Gondar, Gondar, Ethiopia

☯ These authors contributed equally to this work.
* bireswas@gmail.com

**Data Availability Statement:** All relevant data are within the paper and its Supporting Information files.

## Abstract

### Background

Labor pain management is crucial to ensure the quality of obstetric care but it is one of the neglected areas in obstetrics. This study aimed to assess the practice of labor pain management and associated factors among skilled attendants working in public health facilities in Southern, Ethiopia from November 1–January 26, 2019.

### Methods

An Institution-based cross-sectional study design was conducted from November 1–January 26, 2019. A simple random sampling technique was used to select a total of 272 obstetric care providers. Data were collected using pretested, and structured questionnaires. Data were entered to Epi data version 3.1 statistical software and exported to SPSS 22 for analysis. Bivariate and multivariate logistic regression analyses were performed to identify associated factors. P-value <0.05 with 95% confidence level were used to declare statistical significance.

### Result

Overall, 37.5% (95%CI: 32%, 43%) of health care providers had a good practice on non-pharmacological labor pain management. Clinical experience of 5 years and above (AOR = 2.91, 95%CI: 1.60, 5.42), favorable attitude (AOR = 2.82, 95%CI: 1.56, 5.07), midwife profession (AOR = 1.45, 95%CI: 1.98, 4.27), and working in satisfactory delivery rooms (AOR = 3.45, 95%CI: 2.09, 7.43), were significantly associated with a health professional good practice of labor pain management.

**Funding:** The author(s) received no specific funding for this work.

**Competing interests:** The authors have declared that no competing interests exist

## Conclusion

This study showed that the practice of non-pharmacological labor pain management was poor in public health facilities in Gamo and Gofa zone. It was observed that having a favorable attitude, having ≥5 years of work experience, being a midwife by professional, and having a satisfactory delivery room were found to be significant predictors of the practice of non-pharmacological labor pain management. Therefore, all health facilities and concerned bodies need efforts to focus on providing training to midwives on non-pharmacological labor pain management practice.

## Introduction

Labour pain is a universal concern for women. Women's experience of pain during labor varies greatly. Some women feel little pain whilst others find the pain extremely distressing. Addressing pain relief during childbirth is a way of promoting a satisfactory birth experience and a healthy reproductive outcome for women [1–4]. Non-pharmacological labor pain management (NPLPM) methods are non-invasive, cheap, and simple [4–7]. Lack of labor pain management practice is one of the indications of poor quality of care and contributing factors of low utilization of institutional delivery and indirect contributing factors of maternal morbidity and mortality [8, 9]

Globally maternal mortality is unacceptably high. About 295 000 women died during and following pregnancy and childbirth in 2017. The vast majority of these deaths (94%) occurred in low-resource settings, and most could have been prevented. Sub-Saharan Africa and Southern Asia accounted for approximately 86% (254 000) of the estimated global maternal deaths in 2017 [10, 11]. According to EDHS (Ethiopian Demographic and Health Survey), the 2016 report MMR(Maternal Mortality Ratio) in Ethiopia was 412 per 100,000 live birth. Only 26% of births in Ethiopia were delivered at a health facility by a skilled attendant [10]. Studies have shown that the use of non-pharmacological labor pain management during labor is important to improve maternal satisfaction and facilitate mothers' ability for attachment and infant caregiving [12, 13]. Non-pharmacological labor pain management has many advantages like having shorting the progress of labor, minimizing the need for oxytocin and anesthesia, the strength of maternity care user and providers bonding, reducing instrumental deliveries, and decreasing by 50% of cesarean section [14]. Additionally, anxiety, postpartum depression, postpartum bleeding, postpartum fever, a low Apgar score of neonates (<7 at 5 minutes), and prolonged infant hospitalization can be also decreased [15]. The Ethiopian FMoH (Federal Ministry of Health) has developed and implemented the first standard of midwifery care practice in 2013. Among the practice, competencies stated the provision of physical and psychological support and use of non-pharmacological comfort measures during labor and birth are listed as core competencies under practice standard III [16]. This is one of the components to improve the quality of maternal health services. However, its actual practice is not recognized [17, 18]. Therefore, this study has assessed the practice of labor pain management and associated factors among skilled attendants working in public health facilities in the Gamo and Gofa zone. Southern Ethiopia

## Methods

### Study area and period

The study was conducted in selected public health facilities in Gamo and Gofa zone in south Ethiopia, November 1–January 26, 2019

## Study design

A multicenter cross-sectional study was conducted

## Sample size and sampling technique

A simple random sampling technique was used to select two hundred seventy-two (272) sampled obstetric caregivers, (including midwives, nurses, and health officers) who were giving obstetric care in the delivery room.

**Inclusion criteria.**   All obstetric caregivers, (including midwives, nurses, and health officers) who were giving obstetric care in the delivery room were included in the study

## Exclusion criteria

Those health care providers who were absent during data collection due to annual leave

**Measurement.**   Knowledge about labor pain management methods was measured by a 14-item questionnaire adapted from previous studies [3, 19, 20]. The scale for assessing knowledge was from 0 to 14 scores. Correct answers were given a score of one and incorrect answers zero. Those who scored less than the mean value were considered to have poor knowledge while those who scored greater than or equal to the mean value were considered as having good knowledge.

Labor pain management practice was measured by a 9-item checklist adapted from previous studies [3, 19, 20]. The checklist for assessing practice was from zero to nine scores. Having correct practice was given a score of one and no practice were zero. Those who scored less than the mean value were considered to have poor practice while those who scored greater than or equal to the mean value were considered as having good practice

## Data collection tool

A pretested, structured self-administered, and observational checklist questionnaire was prepared based on reviewing relevant literature. The instrument was pretested for its reliability. Qualified obstetricians and public health expertise reviewed the content validity of the questionnaire. The questionnaires were designed in English. All skilled attendants who were working in labor wards of public health institutions found in the Gamo and Gofa zone and who fulfilled eligibility criteria were interviewed by structured questionnaires. Nine data collectors were recruited and principal investigators supervise the data collection process

## Data processing and analysis

Before starting the actual data collection, one-day training was provided for both data collectors and supervisors on objectives and approaches to study subjects. The pretest was conducted 5% of the total sample size outside the study area. The reliability of the questionnaires was checked via SPSS by reliability index measurement for practice questions (Cronbach's alpha) which was 0.85. Data were coded, cleaned, edited, and entered into EPI data version 3.1 and exported to SPSS version 22.0 for statistical analysis. Descriptive statistics with percentages were employed. All variables were analyzed in bivariate logistic regression and those variables having P-value less than 0.25 were entered into multivariable logistic regression analyses. In multivariable logistic regression analyses variables with P-value, less than 0.05 were considered as significant. Hosmer–Lemeshow goodness of fit test was used to check the model fitness. Adjusted odds ratio with 95% confidence interval was used to determine the presence and direction of the association between covariates and the outcome variable

**Ethical approval and consent to participant.**  The study protocol was ethically approved by the Ethical Review Board (IRB) of Arba Minch University College of Medicine and Health Sciences. An official letter was written to each hospital and health center. The study posed a low or no more than minimal risk to the study participants. Also, the study did not involve any invasive procedures. Accordingly, after the objective of the study was explained, verbal informed consent was obtained from all participants

## Results

### Socio-demographic characteristics

272 health professionals participated in the study, yielding a response rate of 96%. The mean age of study participants was 29.9 years old. More than half (57%) of the study participants were females and 155(66.5%) were married. 107(39.3%) of health professionals were orthodox religious followers (**Table 1**).

**Knowledge of health care providers on labor pain management practice.**  Of the total respondents, 223(82.0%) stated back massage can be used to relieve labor pain, and the majority 217(79.8%) of health care providers mentioned relaxation/breathing techniques used to relieve labor pain (**Table 2**). Of the total respondents, 140(51.5%) of health care providers have good knowledge and 132(48.5%) of health care providers have poor knowledge of labor pain management respectively.

**The attitude of health care providers towards labor pain management practice.**  Overall, 195(71.7%) of health care providers have unfavorable attitudes toward labor pain management and 77(28.3%) of health care providers have a favorable attitude. The majority of 263 (96.7%) health care providers believe that analgesia is necessary for managing labor pain and 147(54%) believe that labor pain management reduces postpartum depression. Above the half 197(72%) agree that continuous labor support increase maternal and fetal bonding, besides, 198(72.8%) of respondents believe that spousal presence during labor facilitate delivery

**The practice of non-pharmacological labor pain management.**  Of the total respondents, the majority of 154(56.6%) of health care providers allow companions of her choice during labor and delivery and 184(67.6%) health care providers use psychotherapy to relieve labor pain. The majority of 182(66.9%) and 210(77.2%) of health care providers use relaxation/breathing techniques and massage the back to relieve labor pain respectively (**Table 3**). Overall, 170(62.5%) of health care providers have poor practice and 102(37.5%) have a good practice of non-pharmacological labor pain management

**Factors associated with the practice of non-pharmacological labor pain management.** Binary Logistic regression was performed to assess the association of each independent variable with practice labor pain management. The factors that showed a p-value of less than 0.25 were added to the multivariable regression model. The result revealed that on the Bivariate analysis variables like clinical year of experience, the attitude of health care providers was some of the variables significantly associated with labor pain management practice

In multivariable logistic regression clinical year of experience, having a favorable attitude, having a favorable delivery room, and being midwifery by profession were some of the variables significantly associated with the practice of non-pharmacological labor pain management at a P-value of <0.05.

Respondents with clinical experience of 5 years and above were 2.91 times more likely to perform good practice than respondents with clinical experience of less and equal to two years with (AOR = 2.91, 95%CI: 1.60, 5.42), and respondents who had a favorable attitude towards labor pain management were 2.82 times more likely performing good practice than others with AOR = 2.82, 95%CI: 1.56, 5.07), in addition to this, those health care providers who were

**Table 1. Socio-demographic characteristics of respondents in Gamo and Gofa zone Public health facility southern, Ethiopia.**

| Characteristics | Frequency (N = 272) | Percentage (%) |
|---|---|---|
| **Age** | | |
| 21–25 | 60 | 22.1 |
| 26–30 | 79 | 29.0 |
| 31–35 | 98 | 36.0 |
| > = 36 | 35 | 12.9 |
| **Marital status** | | |
| Married | 181 | 66.5 |
| Divorced | 3 | 1.1 |
| Single | 88 | 32.4 |
| **Sex** | | |
| Male | 117 | 43.0 |
| Female | 155 | 57.0 |
| **Profession** | | |
| Midwife | 173 | 63.6 |
| Nurse | 36 | 13.2 |
| Public health officer | 63 | 23.2 |
| **Highest qualification** | | |
| Masters | 14 | 5.1 |
| Bachelor degree | 154 | 56.6 |
| Diploma | 104 | 38.2 |
| **Religion** | | |
| Orthodox | 107 | 39.3 |
| Protestant | 78 | 28.7 |
| Muslim | 87 | 32.0 |
| **Ethnicity** | | |
| Gamo | 94 | 34.6 |
| Gofa | 80 | 29.4 |
| Wolayta | 11 | 4.0 |
| Amhara | 45 | 16.5 |
| Oromo | 22 | 8.1 |
| Others | 20 | 7.4 |
| **Work experience** | | |
| 1–2 | 90 | 33.1 |
| 3–4 | 88 | 32.4 |
| ≥ 5 | 94 | 34.6 |

working in favorable delivery rooms were 3.45 times more likely to perform good practice than others with (AOR = 3.45, 95%CI: 2.09, 7.43), those health care providers who have midwifery in the profession were 1.45 times more likely to perform good practice than others with (AOR = 1.45, 95%CI: 1.98, 4.27). (**Table 4**)

## Discussion

Pain is more than just a feeling of discomfort that leads to mental health conditions like depression and anxiety. The poor practice may lead to a woman's dissatisfaction and leads to negative birth and it may affect her emotional well-being and willingness to have another

**Table 2. Knowledge of health care providers on labor pain management practice in Gamo and Gofa zone Public health facilities southern, Ethiopia.**

| Characteristics | Frequency (N = 272) | Percentage (%) |
|---|---|---|
| **Which pharmacologic method do you know** | | |
| **Systemic** | | |
| Yes | 181 | 66.5 |
| No | 91 | 33.4 |
| **NSAID drugs** | | |
| Yes | 186 | 68.4 |
| No | 86 | 31.6 |
| **Epidural analgesia** | | |
| Yes | 175 | 64.3 |
| No | 97 | 35.7 |
| **Inhalational** | | |
| Yes | 190 | 69.9 |
| No | 81 | 29.8 |
| **Which non-pharmacologic method do you know** | | |
| **Psychotherapy** | | |
| Yes | 210 | 73.9 |
| No | 71 | 26.1 |
| **Allow companion of her choice** | | |
| Yes | 211 | 77.6 |
| No | 61 | 22.4 |
| **Show the patient how to bear down** | | |
| Yes | 212 | 77.9 |
| No | 60 | 22.1 |
| **Relaxation/breathing technique** | | |
| Yes | 217 | 79.8 |
| No | 55 | 20.2 |
| **Massage the back** | | |
| Yes | 223 | 82.0 |
| No | 49 | 18 |
| **Allow the mother to ambulate** | | |
| Yes | 218 | 80.1 |
| No | 54 | 19.9 |
| **Transcutaneous electrical nerve stimulation** | | |
| Yes | 77 | 28.5 |
| No | 195 | 71.7 |
| **Subcutaneous water injection** | | |
| Yes | 158 | 58.1 |
| No | 114 | 41.9 |
| **Diversional therapy** | | |
| Yes | 162 | 59.6 |
| No | 110 | 40.4 |
| **Hypnosis** | | |
| Yes | 81 | 29.8 |
| No | 191 | 70.2 |
| **Acupuncture** | | |
| Yes | 83 | 30.5 |

(*Continued*)

**Table 2.** (Continued)

| Characteristics | Frequency (N = 272) | Percentage (%) |
|---|---|---|
| **Which pharmacologic method do you know** | | |
| No | 189 | 69.5 |
| **Music therapy** | | |
| Yes | 180 | 66.2 |
| No | 92 | 33.8 |

baby. Using non-pharmacological interventions serves as a way of relaxing the woman in labor and making her more comfortable during the process [21].

The finding of this study disclosed that 37.5% of the study participants had a good practice of non-pharmacological labor pain management. The finding of this study was in line with a study done in the Kembata Tembaro zone, Southern Ethiopia (37.9%), and Amhara region referral hospitals (40.1%), respectively [7, 22]. The finding of this study is lower than the studies conducted in Nigeria (48.4%), and Tigray Region, North Ethiopia (43.4%) [1, 23] respectively. The discrepancy might be due to the difference in health facilities and sample size It was consistent with studies done in Bangladesh and Ghana where allowing laboring women to move freely, showing the patient how to bear down, allowing companionship, and massaging the back was the most applied non-pharmacologic pain relief methods [2, 24]. But it was inconsistent with the study done in Nairobi [21]. The discrepancy might be due to the socio-cultural characteristics of mothers, health facilities rules and regulations, traditional practices at birth, and poor communication of health professionals.

Clinical years of experience, having a favorable attitude, having a satisfactory delivery room, and being a midwife by profession were some of the variables significantly associated with the practice of non-pharmacological labor pain management in multivariable analyses. This finding is similar to another study conducted in Ethiopia, which showed that clinical years of experience, having a favorable attitude, and having a satisfactory delivery room were associated with the good practice of non-pharmacological labor pain management [1, 19]

**Table 3. The practice of non-pharmacological labor pain management in Gamo and Gofa zone Public health facility southern, Ethiopia.**

| S. no | List of cheek list | Observational cheek list to assess practice labor pain management | |
|---|---|---|---|
| | | Yes | No |
| 1. | Health Care providers show the patient how to bear down during labor and delivery | 153 (56.3%) | 119 (43.7%) |
| 2. | Health Care providers allow companion of her choice during labor and delivery | 154 (56.6%) | 118 (43.4%) |
| 3. | Health care providers massage the back to relieve labor pain | 210 (77.2%) | 62(22.8%) |
| 4. | Health care providers use hot or cold pack compress to relieve labor pain | 182 (66.9%) | 90(33.1%) |
| 5. | Health care providers use psychotherapy to relieve labor pain | 184 (67.6%) | 88(32.4%) |
| 6. | Health care providers use relaxation/ breathing techniques to relieve labor pain | 182 (66.9%) | 90(33.1%) |
| 7. | Health care provider allows the mother to ambulate/labor exercise | 158 (58.1%) | 114 (41.9%) |
| 8. | Health care providers use subcutaneous water injection | 2(0.7%) | 270 (99.3%) |

**Table 4. Factors associated with the practice of non-pharmacological labor pain management in Gamo and Gofa zone Public health facilities' southern, Ethiopia.**

| Variables | The practice of labor pain management | | COR with 95% CI | AOR with 95% CI |
|---|---|---|---|---|
| | Good practice | Poor practice | | |
| Year of experiences | | | | |
| 1–2 | 22 | 68 | 1 | 1 |
| 3–4 | 30 | 58 | 1.59(0.83–3.07) | 6.98(2.29–12.5) * |
| ≥ 5 | 50 | 44 | 3.51(1.87–6.57) * | 2.91(1.60–5.42) * |
| Knowledge of health care providers | | | | |
| Poor knowledge | 54 | 78 | 1 | 1 |
| Good knowledge | 48 | 92 | 0.75(0.46–1.23) | 0.50(0.24–1.16) |
| The attitude of health care providers | | | | |
| Unfavorable | 45 | 87 | 1 | 1 |
| Favorable | 32 | 108 | 3.40(1.97–5.89) * | 2.82(1.56–5.07) * |
| Profession | | | | |
| Midwife | 71 | 102 | 1.39(0.76–2.55) | 1.45(1.98–4.27) * |
| Nurse | 10 | 26 | 0.76(0.31–1.88) | 0.98(0.32–2.92) |
| Health officer | 21 | 42 | 1 | 1 |
| Delivery room status | | | | |
| Conducive | 92 | 112 | 0.73 (0.36–1.25) | 3.45(2.09–7.43) * |
| Not conducive | 36 | 32 | 1 | 1 |

Respondents with clinical experience of 5 years and above were 2.91 times more likely to perform good practice than respondents with clinical experience of less and equal to two years. This result was consistent with the study conducted in northern Ethiopia, this could be the reason that the more they stay at work the more they can understand labor pain. Besides the more they stay in a health facility they receive training related to labor pain management and update their knowledge and they will perform more. Respondents who had favorable attitudes towards labor pain management were 2.82 times more likely to perform good practice than others. This finding is similar to other studies conducted in the northern part of Ethiopia [1]. The fact for this is a good attitude of health care professionals is more likely to have the skills and perform the activities in a good manner. Besides, those health care providers who were working in satisfactory delivery rooms were 3.45 times more likely to perform good practice than others. similarly, those health care providers who have midwifery in the profession were 1.45 times more likely to perform good practice than others. The identified factors should be addressed at both institutional and national levels to increase pain management practice for expectant mothers. Providing training for health care providers on non-pharmacological labor pain management and creating a satisfactory delivery room is vital to increases the practice of labor pain management. Besides, it is important for maternity care providers to respect the individuality of each client and be prepared to modify the delivery environment to attract more mothers to the health facility

## Conclusion

This study revealed the poor practice of non-pharmacological labor pain management. Favorable attitude, clinical years of experience, a midwife by profession, and having a satisfactory delivery room were found to be significant predictors of the practice of non-pharmacological labor pain management. The identified factors should be addressed to improve pain

management for expectant mothers. Poor practice in the birth process may lead to negative birth experiences of the mother and it may affect her emotional well-being and willingness to have another baby. However, supporting women in labor and delivery is vital and one of the components of compassionate and respectful maternity care. It should be emphasized that it should be trained to midwives on non-pharmacological labor pain management practices. Future prospective studies should be needed to examine the effect of non-pharmacologic labor pain management methods on labor pain relief and other related maternal and neonatal outcomes.

## Supporting information

**S1 Data.**
(SAV)

## Acknowledgments

The authors would like to thank data collectors and respondents who participated in this study

## Author Contributions

**Conceptualization:** Biresaw Wassihun, Yosef Alemayehu, Teklemariam Gultie, Beemnet Tekabe, Birhaneselasie Gebeyehu.

**Data curation:** Biresaw Wassihun, Yosef Alemayehu, Teklemariam Gultie, Beemnet Tekabe, Birhaneselasie Gebeyehu.

**Formal analysis:** Biresaw Wassihun, Yosef Alemayehu, Teklemariam Gultie.

**Funding acquisition:** Biresaw Wassihun, Yosef Alemayehu, Teklemariam Gultie.

**Investigation:** Biresaw Wassihun, Yosef Alemayehu, Teklemariam Gultie, Beemnet Tekabe, Birhaneselasie Gebeyehu.

**Methodology:** Biresaw Wassihun, Yosef Alemayehu, Teklemariam Gultie, Beemnet Tekabe, Birhaneselasie Gebeyehu.

**Project administration:** Biresaw Wassihun, Yosef Alemayehu, Teklemariam Gultie.

**Resources:** Biresaw Wassihun, Yosef Alemayehu, Teklemariam Gultie, Beemnet Tekabe, Birhaneselasie Gebeyehu.

**Software:** Biresaw Wassihun, Yosef Alemayehu, Teklemariam Gultie, Birhaneselasie Gebeyehu.

**Supervision:** Biresaw Wassihun, Yosef Alemayehu, Teklemariam Gultie, Beemnet Tekabe, Birhaneselasie Gebeyehu.

**Validation:** Biresaw Wassihun, Yosef Alemayehu, Teklemariam Gultie.

**Visualization:** Biresaw Wassihun, Yosef Alemayehu, Teklemariam Gultie, Beemnet Tekabe, Birhaneselasie Gebeyehu.

**Writing – original draft:** Biresaw Wassihun, Yosef Alemayehu, Teklemariam Gultie, Birhaneselasie Gebeyehu.

**Writing – review & editing:** Biresaw Wassihun, Yosef Alemayehu, Teklemariam Gultie, Beemnet Tekabe, Birhaneselasie Gebeyehu.

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
