## [Decision Letter · Decision Letter 0]

11 May 2020

PONE-D-20-00957

Non-Pharmacological Labor Pain Management Practice and Associated Factors among Skilled Attendants working in public health facilities in Gamo and Gofa zone. Southern Ethiopia

PLOS ONE

Dear Dr. Alemu,

Thank you for submitting your manuscript to PLOS ONE. After careful consideration, we feel that it has merit but does not fully meet PLOS ONE’s publication criteria as it currently stands. Therefore, we invite you to submit a revised version of the manuscript that addresses the points raised during the review process.

We would appreciate receiving your revised manuscript by Jun 25 2020 11:59PM. To enhance the reproducibility of your results, we recommend that if applicable you deposit your laboratory protocols in protocols.io, where a protocol can be assigned its own identifier (DOI) such that it can be cited independently in the future. For instructions see: http://journals.plos.org/plosone/s/submission-guidelines#loc-laboratory-protocols

We look forward to receiving your revised manuscript.

Kind regards,

Nülüfer Erbil, Ph.D, Prof.

Academic Editor

PLOS ONE

Journal Requirements:

4. Your ethics statement must appear in the Methods section of your manuscript. If your ethics statement is written in any section besides the Methods, please move it to the Methods section and delete it from any other section. Please also ensure that your ethics statement is included in your manuscript, as the ethics section of your online submission will not be published alongside your manuscript.

Reviewers' comments:

Reviewer's Responses to Questions

**Comments to the Author**

1. Is the manuscript technically sound, and do the data support the conclusions?

Reviewer #1: Yes

Reviewer #2: Yes

2. Has the statistical analysis been performed appropriately and rigorously? 

Reviewer #1: No

Reviewer #2: Yes

3. Have the authors made all data underlying the findings in their manuscript fully available?

Reviewer #1: Yes

Reviewer #2: Yes

4. Is the manuscript presented in an intelligible fashion and written in standard English?

Reviewer #1: Yes

Reviewer #2: No

5. Review Comments to the Author

Reviewer #1: Thank you for your project on this important topics, I have attached my comments int he file, but my big concern about the instrument that you used, no information is given about the instrument how you measured knowledge attitute of providers..

who develop them, validity, reliability ?

My all comments are in the pdf file.

Reviewer #2: General.

Manuscript should be edited by native spekaer.

There are too many grammatical errors in the manuscript. Please correct!

There are many spelling mistakes in the manuscript . Please correct!

For Results:

Is religion of the health professionals effect on good or poor practice of labor pain management? Please briefly comment with a sentence!

Two yes are written in the last line in table 3 (on Acupuncture). One of them should be no.

Discussion:

The discussion was written very poorly. This section should be improved with new references

It should be emphasized the new perspectives in the discussion section in the light of your results

For example: Poor practice in birth process may lead to negative birth experiences of mother and traumatic perception of birth .

It should be emphasized the relationship between good or poor practice and also birth center, birth enviroment, birth team,

It should be given skill training for health care providers on nonpharmacological applications in relieving birth pain.

Suggestioned some references:

Aktaş, S., & Aydın, R. (2019). The analysis of negative birth experiences of mothers: a qualitative study. Journal of reproductive and infant psychology, 37(2), 176-192.

Guise, J. M., & Segel, S. (2008). Teamwork in obstetric critical care. Best practice & research Clinical obstetrics & gynaecology, 22(5), 937-951.

Oosthuizen, S. J., Bergh, A. M., Grimbeek, J., & Pattinson, R. C. (2019). Midwife-led obstetric units working ‘CLEVER’: Improving perinatal outcome indicators in a South African health district. South African Medical Journal, 109(2), 95-101.

Conclusion:

It should be emphasized that ıt should be trained to midwives on non-pharmacological labor pain management practice.

6. PLOS authors have the option to publish the peer review history of their article (what does this mean?). If published, this will include your full peer review and any attached files.

Reviewer #1: No

Reviewer #2: No

---

## [Author Response · Author response to Decision Letter 0]

1 Jun 2020

Thank you very much we have addressed All comments and suggestions of both reviewers and we have attached response thanks

---

## [Decision Letter · Decision Letter 1]

4 Aug 2020

PONE-D-20-00957R1

Non-Pharmacological Labor Pain Management Practice and Associated Factors among Skilled Attendants working in public health facilities in Gamo and Gofa zone, Southern Ethiopia: A cross-sectional study

PLOS ONE

Dear Dr. Alemu,

Thank you for submitting your manuscript to PLOS ONE. After careful consideration, we feel that it has merit but does not fully meet PLOS ONE’s publication criteria as it currently stands. Therefore, we invite you to submit a revised version of the manuscript that addresses the points raised during the review process.

We look forward to receiving your revised manuscript.

Kind regards,

Nülüfer Erbil, Ph.D, Prof.

Academic Editor

PLOS ONE

Reviewers' comments:

Reviewer's Responses to Questions

**Comments to the Author**

1. If the authors have adequately addressed your comments raised in a previous round of review and you feel that this manuscript is now acceptable for publication, you may indicate that here to bypass the “Comments to the Author” section, enter your conflict of interest statement in the “Confidential to Editor” section, and submit your "Accept" recommendation.

Reviewer #2: All comments have been addressed

Reviewer #3: All comments have been addressed

2. Is the manuscript technically sound, and do the data support the conclusions?

Reviewer #2: Yes

Reviewer #3: Partly

3. Has the statistical analysis been performed appropriately and rigorously? 

Reviewer #2: Yes

Reviewer #3: Yes

4. Have the authors made all data underlying the findings in their manuscript fully available?

Reviewer #2: Yes

Reviewer #3: Yes

5. Is the manuscript presented in an intelligible fashion and written in standard English?

Reviewer #2: Yes

Reviewer #3: No

6. Review Comments to the Author

Reviewer #2: The authors mad the corrections given. This study has the ability to raise awareness about good practices, especially in developing countries. For me, ıt is suitable for publication

Reviewer #3: The manuscript has been improved much. However, I do have some major comments.

1. There are still many grammar and spelling errors in the article. For example, “Associated factors among skilled” in the abstract, the first letter should be lower case. “in Southern, Ethiopia”, please add a period at the end of the sentence. Same errors exist in other parts of the manuscript, please carefully check the manuscript and correct these errors.

2. Introduction section, the authors introduced “labor pain management” and its role in the first paragraph. However, the transition to maternal mortality in the second paragraph seems illogical. “labor pain management” is not a key factor of maternal mortality.

3. Methods section, the content is too much and should be reduced. There are some improper English expressions. This research is indeed a facility-based investigation, no need to describe too much the “study area and period” or population.

4. Results section, just summarize the key findings, 3 or 4 tables are enough. The authors used logistic regression model to explore factors associated with good labor pain management. Score is a continuous variable, are the results same if using linear regression model? Does the score vary by facilities or by regions (Gamo vs Gofa)?

5. Discussion section, besides discussing the current status and associated factors of “non-pharmacological labor pain management” among skilled attendants in public health facilities, the authors should address how to improve the practice based on current findings and the guide released by government.

7. PLOS authors have the option to publish the peer review history of their article (what does this mean?). If published, this will include your full peer review and any attached files.

Reviewer #2: No

Reviewer #3: No

---

## [Author Response · Author response to Decision Letter 1]

7 Aug 2020

Author’s Point-by-Point Response to the Reviewer's and Editors Reports

Title: Non-Pharmacological Labor Pain Management Practice and Associated Factors among Skilled Attendants working in public health facilities in Gamo and Gofa zone, Southern Ethiopia: A cross-sectional study 

Corresponding author: Biresaw Wassihun /bireswas@gmail.com

Authors

1. Biresaw Wassihun

2. Yosef Alemayehu

Manscurpuit number: PONE-D-20-00957

Journal: PloS one

Article type: Research article

Point by point response to Reviewers and Editors

First of all, the authors would like to thank Plos one Journal editors and the respective reviewers for reviewing our manuscript and providing the necessary comments to be corrected. As per the comments given, we have made corrections point by point to comment. The authors tried to answer all the issues raised by the editorial team and reviewers. Thank you again providing this interesting and constrictive comment 

Point by point response to Editor

Dear Nülüfer Erbil, Ph.D., Prof

1. Please ensure that your manuscript meets PLOS ONE's style requirements 

Response: Thank you very much we had applied journal requirement 

2. Thank you for submitting your manuscript to PLOS ONE. After careful consideration, we feel that it has merit but does not fully meet PLOS ONE’s publication criteria as it currently stands. Therefore, we invite you to submit a revised version of the manuscript that addresses the points raised during the review process3. Please amend either the title on the online submission form (via Edit Submission) or the title in the manuscript so that they are identical 

Response: Dear respected Editor Nülüfer Erbil, Ph.D., Prof thank you for providing interesting and constrictive comments so, we will provide each separately, thank you again 

Point by point response to Reviewer# 3

Reviewer #2: The authors mad the corrections are given. This study can raise awareness about good practices, especially in developing countries. For me, ıt is suitable for publication.

Response: We would like to say thank you very much for your invaluable comments and suggestions. We considered and modified and rewrote again based on your constructive issues regarding language, coherence, and comprehensibility of the manuscript 

 Reviewer #3: The manuscript has been improved much. However, I do have some major comments.

Question 1. There are still many grammar and spelling errors in the article. For example, “Associated factors among skilled” in the abstract, the first letter should be lower case. “in Southern, Ethiopia”, please add a period at the end of the sentence. Same errors exist in other parts of the manuscript, please carefully check the manuscript and correct these errors

Response 1: We would like to say thank you very much for your invaluable comments and suggestions. We considered, modified, and rewrote again based on your constructive issues regarding language, and grammar. The abstract was corrected as follows 

The health and well-being of a mother and child at birth largely determine the future health and wellness of the entire family. Labor pain management is crucial to ensure the quality of obstetric care but it is one of the neglected areas in obstetrics. Therefore, this study aimed to assess the practice of labor pain management and associated factors among skilled attendants working in public health facilities in Southern, Ethiopia from November 1–January 26, 2019 

Question 2. Introduction section, the authors introduced “labor pain management” and its role in the first paragraph. However, the transition to maternal mortality in the second paragraph seems illogical. “Labor pain management” is not a key factor of maternal mortality

Response 2: Thank you very much. Dear respected reviewer as you know that one of the factors that affect delivery in public health institutions is the lack of health care providers' support and inadequate quality that is related to pain relief. Based on literature Lack of labor pain management practice is one of the unspoken causes of low utilization of institutional delivery so if there is low utilization of institutional delivery there is a high number of home delivery. Home delivery itself as its contribution to maternal mortality and in Ethiopia majority 74% of mothers gave birth in the home so lack of continuous labor support or lack of compassionate and respectful maternity care will be one of the indirect causes of maternal mortality and morbidity.

Finally, thank you again and we will as modify based on your comment and suggestion 

Question 3. In the methods section, the content is too much and should be reduced. There are some improper English expressions. This research is indeed a facility-based investigation, no need to describe too much the “study area and period” or population

 Response 3: It was corrected according to your suggestion, as follows

The study was conducted in selected public health facilities in Gamo and Gofa zone in south Ethiopia, from November 1–January 26, 2019

The source population study area and period was corrected accordingly thank you very much

Question 4. Results section, just summarize the key findings, 3 or four tables are enough.

Response 4. It was corrected according to your suggestion and the number of tables was reduced 

Question 5 . Does the score vary by facilities or by regions (Gamo vs Gofa)?

 Response5 : As a suggestion, we will consider it in the future to assess the disparity of using non-pharmacological labor pain management between health facilities taking health facility as one of the independent variable even including other private health facility

Thanks for your advance 

Question 6. In the discussion section, besides discussing the status and associated factors of “non-pharmacological labor pain management” among skilled attendants in public health facilities, the authors should address how to improve the practice based on current findings and the guide released by the government.

Response 6. The correction was made thanks in-depth, for your nice comment 

The identified factors should be addressed at both institutional and national levels to increase pain management practice for expectant mothers. Providing training for health care providers on non-pharmacological labor pain management and crating satisfactory delivery room is vital to increases practice of labor pain management. In addition, it is important for maternity care providers to respect the individuality of each client and be prepared to modify the delivery environment to attract more mothers to the health facility

---

## [Decision Letter · Decision Letter 2]

15 Oct 2020

PONE-D-20-00957R2

Non-Pharmacological Labor Pain Management Practice and Associated Factors among Skilled Attendants working in public health facilities in Gamo and Gofa zone, Southern Ethiopia: A cross-sectional study

PLOS ONE

Dear Dr. Alemu,

Thank you for submitting your manuscript to PLOS ONE. After careful consideration, we feel that it has merit but does not fully meet PLOS ONE’s publication criteria as it currently stands. Therefore, we invite you to submit a revised version of the manuscript that addresses the points raised during the review process.

We look forward to receiving your revised manuscript.

Kind regards,

Nülüfer Erbil, Ph.D, Prof.

Academic Editor

PLOS ONE

Reviewers' comments:

Reviewer's Responses to Questions

**Comments to the Author**

1. If the authors have adequately addressed your comments raised in a previous round of review and you feel that this manuscript is now acceptable for publication, you may indicate that here to bypass the “Comments to the Author” section, enter your conflict of interest statement in the “Confidential to Editor” section, and submit your "Accept" recommendation.

Reviewer #4: All comments have been addressed

2. Is the manuscript technically sound, and do the data support the conclusions?

Reviewer #4: Yes

3. Has the statistical analysis been performed appropriately and rigorously? 

Reviewer #4: Yes

4. Have the authors made all data underlying the findings in their manuscript fully available?

Reviewer #4: Yes

5. Is the manuscript presented in an intelligible fashion and written in standard English?

Reviewer #4: Yes

6. Review Comments to the Author

Reviewer #4: Thank you for the opportunity to review this manuscript under title Non-Pharmacological Labor Pain Management Practice and Associated Factors among Skilled Attendants working in public health facilities in Gamo and Gofa zone, Southern Ethiopia: A cross-sectional study.

The manuscript has been improved so much. However, I do have some minor comments.

There are still many grammar and spelling and style errors in the manuscript.

Abstract section, in the key words changing the word of Labour to labor its better to write the term in manuscript in the same style (unified).

Introduction section, According to EDHS 2016 report MMR in Ethiopia was 412 per 100,000 live birth.

The EDHS abbreviation of what? al so the MMR & FMoH in the paragraph two line 14.

In the end of the introduction section put full stop point (.)

Methods section, put full stop point (.) at the end of each paragraph for example:

A simple random sampling technique was used to select two hundred seventy-two (272) sampled

obstetric caregivers; (including midwives, nurses, and health officers) who were giving obstetric

care in the delivery room.

- Sample size and sampling technique, mention clearly the inclusion and exclusion criteria.

- In the Measurement part, Channing the word of Labour to labor in the line 6 also changing word of the cheek list to check list in the same line.

- In the Ethical approval and consent to participant, write down the number and date of the ethical approve issue

Results section, in general, revision the results interpretation

- Can chinning the (In this study, 272 health professionals) to (272 health professionals participated in the study, yielding a response rate of)

Discussion section, in general, need revision the discussion part

References section, check the number and the sequence of references in the text with the list of the references for example check the numbers 2, 5, 6, sequence.

7. PLOS authors have the option to publish the peer review history of their article (what does this mean?). If published, this will include your full peer review and any attached files.

Reviewer #4: **Yes: **Warda Hassan Abdullah

---

## [Author Response · Author response to Decision Letter 2]

28 Oct 2020

All of the comments was corrected 

Author’s Point-by-Point Response to the Reviewer's and Editors Reports

Title: Non-Pharmacological Labor Pain Management Practice and Associated Factors among Skilled Attendants working in public health facilities in Gamo and Gofa zone, Southern Ethiopia: A cross-sectional study 

Corresponding author: Biresaw Wassihun /bireswas@gmail.com

Authors

1. Biresaw Wassihun

2. Yosef Alemayehu

3. Teklemariam Gultie

Manscurpuit number: PONE-D-20-00957

Journal: PloS one

Article type: Research article

Point by point response to Reviewers and Editors

First of all, the authors would like to thank Plos one Journal editors and the respective reviewers for reviewing our manuscript and providing the necessary comments to be corrected. As per the comments given, we have made corrections point by point to comment. The authors tried to answer all the issues raised by the editorial team and reviewers. Thank you again providing this interesting and constrictive comment 

Point by point response to Editor

1. Please ensure that your manuscript meets PLOS ONE's style requirements 

Response: Thank you very much we had applied journal requirement 

2. Thank you for submitting your manuscript to PLOS ONE. After careful consideration, we feel that it has merit but does not fully meet PLOS ONE’s publication criteria as it currently stands. Therefore, we invite you to submit a revised version of the manuscript that addresses the points raised during the review process

Response: Dear respected Editor thank you for providing interesting and constrictive comments so, we will provide each separately, thank you again 

Point by point response to Reviewer# 4

Reviewer #4: Thank you for the opportunity to review this manuscript under title Non-Pharmacological Labor Pain Management Practice and Associated Factors among Skilled Attendants working in public health facilities in Gamo and Gofa zone, Southern Ethiopia: A cross-sectional study.

The manuscript has been improved so much. However, I do have some minor comments.

There are still many grammar and spelling and style errors in the manuscript..

Response: We would like to say thank you very much for your invaluable comments and suggestions. We considered and modified and rewrote again based on your constructive issues regarding language, coherence, and comprehensibility of the manuscript 

 Question 1. There are still many grammar and spelling errors in the article. For example” Abstract section, in the key words changing the word of Labour to labor its better to write the term in manuscript in the same style (unified).

Response 1: We would like to say thank you very much for your invaluable comments and suggestions. We considered, modified, and rewrote again based on your constructive issues regarding language, and grammar. The abstract was corrected 

Question 2. Introduction section, According to EDHS 2016 report MMR in Ethiopia was 412 per 100,000 live birth.The EDHS abbreviation of what? al so the MMR & FMoH in the paragraph two line 14.

Response 2: Thank you very much. Dear respected reviewer it was corected accordingly 

Question 3. In the end of the introduction section put full stop point (.)

 Response 3: It was corrected according to your suggestion

Question 4. Methods section, put full stop point (.) at the end of each paragraph for example:

A simple random sampling technique was used to select two hundred seventy-two (272) sampled

obstetric caregivers; (including midwives, nurses, and health officers) who were giving obstetric

care in the delivery room.

Response 4. It was corrected according to your suggestion 

Question 5 . Sample size and sampling technique, mention clearly the inclusion and exclusion criteria.

- In the Measurement part, Channing the word of Labour to labor in the line 6 also changing word of the cheek list to check list in the same line. 

Response5 : Thanks for your advance 

Question 6. Results section, in general, revision the results interpretation

- Can chinning the (In this study, 272 health professionals) to (272 health professionals participated in the study, yielding a response rate of).

Response 6. The correction was made thanks in-depth, for your nice comment 

Question 7 Discussion section, in general, need revision the discussion part

Response7 : Thanks for your advance 

Question 8. References section, check the number and the sequence of references in the text with the list of the references for example check the numbers 2, 5, 6, sequence 

Response8. we have made correction on it 

Question 9 Sample size and sampling technique, mention clearly the inclusion and exclusion criteria

Response9. we have made correction on it as followes 

Inclusion criteria 

All obstetric caregivers; (including midwives, nurses, and health officers) who were giving obstetric care in the delivery room were inclideded in the study

Exclusion criteria 

Those health care provider Absent during data collection due to annual leave

---

## [Decision Letter · Decision Letter 3]

18 Jan 2021

PONE-D-20-00957R3

Non-Pharmacological Labor Pain Management Practice and Associated Factors among Skilled Attendants working in public health facilities in Gamo and Gofa zone, Southern Ethiopia: A cross-sectional study

PLOS ONE

Dear Dr. Alemu,

Thank you for submitting your manuscript to PLOS ONE. After careful consideration, we feel that it has merit but does not fully meet PLOS ONE’s publication criteria as it currently stands. Therefore, we invite you to submit a revised version of the manuscript that addresses the points raised during the review process.

We look forward to receiving your revised manuscript.

Kind regards,

Nülüfer Erbil, Ph.D, Prof.

Academic Editor

PLOS ONE

Reviewers' comments:

Reviewer's Responses to Questions

**Comments to the Author**

1. If the authors have adequately addressed your comments raised in a previous round of review and you feel that this manuscript is now acceptable for publication, you may indicate that here to bypass the “Comments to the Author” section, enter your conflict of interest statement in the “Confidential to Editor” section, and submit your "Accept" recommendation.

Reviewer #3: All comments have been addressed

2. Is the manuscript technically sound, and do the data support the conclusions?

Reviewer #3: Partly

3. Has the statistical analysis been performed appropriately and rigorously? 

Reviewer #3: No

4. Have the authors made all data underlying the findings in their manuscript fully available?

Reviewer #3: Yes

5. Is the manuscript presented in an intelligible fashion and written in standard English?

Reviewer #3: No

6. Review Comments to the Author

Reviewer #3: The authors performed a cross-sectional study to assess the practice of labor pain management and associated factors among skilled attendants working in public health facilities in Southern, Ethiopia. They found that the practice of non-pharmacological labor pain management was poor. They also observed several risk factors reported in previous studies such as a favorable attitude, at least 5 years of work experience, having a satisfactory delivery room, etc. The findings should be of interest to maternity care providers and public health workers, especially in the underdeveloped areas. Although the manuscript has been improved, I still have some comments.

1. Introduction section. Labor pain management practices include pharmacological and non-pharmacological methods. The author should address the current research status of non-pharmacological pain management when introducing the importance of labor pain management.

2. P17 Table 4, “AOR(≥5 years of experience)=7.0(1.60-3.05)” and “AOR(midwife)=7.13(1.06-4.77)”, I am wondering why the confidence intervals did not include the point estimates. The authors should double check the data and verify the results.

3. In the results section, logistic analysis showed that knowledge was not statistically associated with practice. But in the discussion part, the authors ascribed differences in the practice between this study and others to the differences in knowledge. It is reasonable according to the common sense of KAP theory，but it is not supported by the results of this study. The authors should address it carefully and describe it more appropriately.

4. There are still many grammar and spelling errors in the manuscript.

(1) The introduction part, paragraph 2, line 9, “......facilitate mothers' ability for attachment and infant car giving”. It could be “infant care giving”.

(2) In the measurement part, the criteria for evaluating “attitude” was missing.

(3) In the “data processing and analysis”, line 7, the criteria for selecting variables into logistic regression is p < 0.2, while the corresponding description in the results part is < 0.25 (the third row below Table 3).

(4) The title of the Table 2 was “knowledge of non-pharmacological labor pain management practice in Gamo and Gofa zone Public health facilities southern, Ethiopia”. However, the table contains pharmacological and non-pharmacological items. Please use a suitable title instead.

(5) In the Table 4, please give the full spelling of “HCP” in the footnote.

7. PLOS authors have the option to publish the peer review history of their article (what does this mean?). If published, this will include your full peer review and any attached files.

Reviewer #3: No

---

## [Author Response · Author response to Decision Letter 3]

5 Feb 2021

Author’s Point-by-Point Response to the Reviewer's and Editors Reports

Title: Non-Pharmacological Labor Pain Management Practice and Associated Factors among Skilled Attendants working in public health facilities in Gamo and Gofa zone, Southern Ethiopia: A cross-sectional study 

Corresponding author: Biresaw Wassihun /bireswas@gmail.com

Authors

1. Biresaw Wassihun

2. Yosef Alemayehu

3. Teklemariam Gultie

Manscurpuit number: PONE-D-20-00957

Journal: PloS one

Article type: Research article

Point by point response to Reviewers and Editors

First of all, the authors would like to thank Plos one Journal editor, and the respective reviewers for reviewing our manuscript and providing the necessary comments to be corrected. As per the comments given, we have made corrections point by point to comment. The authors tried to answer all the issues raised by the editorial team and reviewers. Thank you again for providing this interesting and constrictive comment 

Point by point response to Editor

1. Please ensure that your manuscript meets PLOS ONE's style requirements 

Response: Thank you very much we had applied the journal requirement 

2. Thank you for submitting your manuscript to PLOS ONE. After careful consideration, we feel that it has merit but does not fully meet PLOS ONE’s publication criteria as it currently stands. Therefore, we invite you to submit a revised version of the manuscript that addresses the points raised during the review process

Response: Dear respected Editor thank you for providing interesting and constrictive comments so, we will provide each separately, thank you again 

Point by point response to Reviewer# 3

Reviewer #3: Thank you for the opportunity to review this manuscript under the title non-Pharmacological Labor Pain Management Practice and Associated Factors among Skilled Attendants working in public health facilities in Gamo and Gofa zone, Southern Ethiopia: 

A cross-sectional study.

The manuscript has been improved so much. However, I do have some minor comments.

There are still many grammar and spelling and style errors in the manuscript.

Response: We would like to say thank you very much for your invaluable comments and suggestions. We considered and modified and rewrote again based on your constructive issues regarding the language, coherence, and comprehensibility of the manuscript 

 Question 1. Introduction section. Labor pain management practices include pharmacological and non-pharmacological methods. The author should address the current research status of non-pharmacological pain management when introducing the importance of labor pain management)

Response 1: We would like to say thank you very much for your invaluable comments and suggestions. We considered, modified, and rewrote again based on your constructive comments

Question 2. 2. P17 Table 4, “AOR(≥5 years of experience)=7.0(1.60-3.05)” and “AOR(midwife)=7.13(1.06-4.77)”, I am wondering why the confidence intervals did not include the point estimates. The authors should double check the data and verify the results.

Response 2: Thank you very much. Dear respected reviewer, it was corrected accordingly 

Question 3. In the results section, logistic analysis showed that knowledge was not statistically associated with practice. But in the discussion part, the authors ascribed differences in the practice between this study and others to the differences in knowledge. It is reasonable according to the common sense of KAP theory，but it is not supported by the results of this study. The authors should address it carefully and describe it more appropriately

 Response 3: It was corrected according 

Question 4. here are still many grammar and spelling errors in the manuscript.

(1) The introduction part, paragraph 2, line 9, “......facilitate mothers' ability for attachment and infant car giving”. It could be “infant care giving 

Response 4. It was corrected according to your suggestion 

Question 5 . In the “data processing and analysis”, line 7, the criteria for selecting variables into logistic regression is p < 0.2, while the corresponding description in the results part is < 0.25 (the third row below Table 3). 

Response5 : Thanks for your advance 

Question 6. The title of the Table 2 was “knowledge of non-pharmacological labor pain management practice in Gamo and Gofa zone Public health facilities southern, Ethiopia”. However, the table contains pharmacological and non-pharmacological items. Please use a suitable title instead.

Response 6. The correction was made thanks in-depth, for your nice comment 

Question 7 In the Table 4, please give the full spelling of “HCP” in the footnote.

Response 7. we have made correction on it

---

## [Decision Letter · Decision Letter 4]

7 Apr 2021

PONE-D-20-00957R4

Non-Pharmacological Labor Pain Management Practice and Associated Factors among Skilled Attendants working in public health facilities in Gamo and Gofa zone, Southern Ethiopia: A cross-sectional study

PLOS ONE

Dear Dr. Alemu,

Thank you for submitting your manuscript to PLOS ONE. After careful consideration, we feel that it has merit but does not fully meet PLOS ONE’s publication criteria as it currently stands. Therefore, we invite you to submit a revised version of the manuscript that addresses the points raised during the review process.

We look forward to receiving your revised manuscript.

Kind regards,

Nülüfer Erbil, Ph.D, Prof.

Academic Editor

PLOS ONE

Journal Requirements:

Reviewers' comments:

Reviewer's Responses to Questions

**Comments to the Author**

1. If the authors have adequately addressed your comments raised in a previous round of review and you feel that this manuscript is now acceptable for publication, you may indicate that here to bypass the “Comments to the Author” section, enter your conflict of interest statement in the “Confidential to Editor” section, and submit your "Accept" recommendation.

Reviewer #3: All comments have been addressed

2. Is the manuscript technically sound, and do the data support the conclusions?

Reviewer #3: Yes

3. Has the statistical analysis been performed appropriately and rigorously? 

Reviewer #3: Yes

4. Have the authors made all data underlying the findings in their manuscript fully available?

Reviewer #3: Yes

5. Is the manuscript presented in an intelligible fashion and written in standard English?

Reviewer #3: Yes

6. Review Comments to the Author

Reviewer #3: The manuscript has been improved. However, I do have some minor comments.

1. P9, abstract, conclusion, line 3: the mentioned factors were significant predictors, not a predictor.

2. P11, measurement, line 6, “labor pain” , the first letter should be uppercased. Line 9, “ ...less than the mean value were considered to have good practice...”. It could be “poor practice”.

3. P12, Data collection tool part, line 3, “Nine data collectors were recruited principal investigators supervise the data collection process”. There should be an “and” between “recruited” and “principal investigators”.

4. P16, Result section. The sex of health care provider was associated with labor pain management practice. But no data presented in the Table 4.

5. P17 Table 4. Please add a footnote for “*”.

6. Discussion section. Last time I mentioned that according to the common sense of KAP theory, knowledge was an influencing factor on practice. But it is not supported by the result of this study. The authors should explain the reasons for the separation of knowledge and practice.

7. There are still some grammar or style errors in the manuscript. For example, many sentences lack a full stop ( “.”). The authors should carefully check the documents to ensure that they meet the journal's format requirements.

7. PLOS authors have the option to publish the peer review history of their article (what does this mean?). If published, this will include your full peer review and any attached files.

Reviewer #3: No

---

## [Author Response · Author response to Decision Letter 4]

21 May 2021

All of the comments was corrected accordingly based on both reviewers and editors comment and suggestion and we have attached point by point response to reviewers and editors

---

## [Decision Letter · Decision Letter 5]

5 Jul 2021

PONE-D-20-00957R5

Non-Pharmacological Labor Pain Management Practice and Associated Factors among Skilled Attendants working in public health facilities in Gamo and Gofa zone, Southern Ethiopia: A cross-sectional study

PLOS ONE

Dear Dr. Alemu,

Thank you for submitting your manuscript to PLOS ONE. After careful consideration, we feel that it has merit but does not fully meet PLOS ONE’s publication criteria as it currently stands. Therefore, we invite you to submit a revised version of the manuscript that addresses the points raised during the review process.

We look forward to receiving your revised manuscript.

Kind regards,

Nülüfer Erbil, Ph.D, Prof.

Academic Editor

PLOS ONE

Journal Requirements:

Reviewers' comments:

Reviewer's Responses to Questions

**Comments to the Author**

1. If the authors have adequately addressed your comments raised in a previous round of review and you feel that this manuscript is now acceptable for publication, you may indicate that here to bypass the “Comments to the Author” section, enter your conflict of interest statement in the “Confidential to Editor” section, and submit your "Accept" recommendation.

Reviewer #3: All comments have been addressed

2. Is the manuscript technically sound, and do the data support the conclusions?

Reviewer #3: Yes

3. Has the statistical analysis been performed appropriately and rigorously? 

Reviewer #3: Yes

4. Have the authors made all data underlying the findings in their manuscript fully available?

Reviewer #3: Yes

5. Is the manuscript presented in an intelligible fashion and written in standard English?

Reviewer #3: Yes

6. Review Comments to the Author

Reviewer #3: The manuscript has been improved. However, there are still some minor errors that need to be corrected.

Question 1. P11, measurement, line 6, “labor pain”, the first letter should be uppercase.

Question 2. P17. The footnote you added to the Table 4, “statical significance at p-value less than 0.05”, spelling mistake for the word "statistical". We know that the table present the 95% confidence intervals, so the statistical level for α is 0.05. The CI of OR does not include 1, indicating that the P value is less than 0.05. If the asterisks in the table have special meaning other than the P value, a footnote should be added. Otherwise, it’s not necessary.

Question 3. Please use punctuation correctly. There are still many sentences lacking a full stop (“.”).

7. PLOS authors have the option to publish the peer review history of their article (what does this mean?). If published, this will include your full peer review and any attached files.

Reviewer #3: No

---

## [Author Response · Author response to Decision Letter 5]

27 Jul 2021

All of the comments was corrected

---

## [Decision Letter · Decision Letter 6]

3 Sep 2021

PONE-D-20-00957R6Non-Pharmacological Labor Pain Management Practice and Associated Factors among Skilled Attendants working in public health facilities in Gamo and Gofa zone, Southern Ethiopia: A cross-sectional study

PLOS ONE

Dear Dr. Alemu,

Thank you for submitting your manuscript to PLOS ONE. After careful consideration, we feel that it has merit but does not fully meet PLOS ONE’s publication criteria as it currently stands. Therefore, we invite you to submit a revised version of the manuscript that addresses the points raised during the review process.

We look forward to receiving your revised manuscript.

Kind regards,

Nülüfer Erbil, Ph.D, Prof.

Academic Editor

PLOS ONE

Journal Requirements:

Reviewers' comments:

Reviewer's Responses to Questions

**Comments to the Author**

1. If the authors have adequately addressed your comments raised in a previous round of review and you feel that this manuscript is now acceptable for publication, you may indicate that here to bypass the “Comments to the Author” section, enter your conflict of interest statement in the “Confidential to Editor” section, and submit your "Accept" recommendation.

Reviewer #3: All comments have been addressed

2. Is the manuscript technically sound, and do the data support the conclusions?

Reviewer #3: Yes

3. Has the statistical analysis been performed appropriately and rigorously? 

Reviewer #3: Yes

4. Have the authors made all data underlying the findings in their manuscript fully available?

Reviewer #3: Yes

5. Is the manuscript presented in an intelligible fashion and written in standard English?

Reviewer #3: No

6. Review Comments to the Author

Reviewer #3: There are still some errors in the manuscript. The authors must ensure that the grammar, format, and punctuation meet the requirements of the journal.

1.Abstract, Methods, “Institution-based cross-sectional study design was conducted from November 1– January 26, 2019”, It is better to be expressed as “An institution-based cross-sectional study was conducted……”.

2.Abstract, Results, “37.5% (95%CI: 32, 43%) of health care providers……” , it should be “37.5% (95%CI: 32%, 43%). Please pay attention to that the AOR should be described as “(AOR=2.82, 95%CI: 1.56, 5.07)”.

3.Abstract, Conclusion, “It was observed that; having a favorable attitude, having ≥5 years of work experience, …...”. Please remove the “;” from the sentence.

4. Introduction, paragraph 2, line 7, “Studies have been shown that ......”, It should be “Studies have shown that.....”. Line 13, “Additionally, Anxiety, postpartum depression.......”, the first letter of “Anxiety” should be lower-case.

5. Methods, Exclusion criteria, “Those health care providers absent.....”, It should be “Those health care providers who were absent.....”.

6. Results, paragraph 1, line 2, The “SD” in “29.9 ± SD4.89” should be removed.

7. It is better to describe the results using past tense sentences.

8. Table 3, please use the percent sign correctly. And in the sentences that begin with “Health Care providers.....”, “Care” should be lower-case.

9. P16, Table 4, Please use correct table format. Row 5, collumn4, “1,87” should be “1.87”.

10. Except for the comments above, there were still many grammar and style errors in the manuscript. The authors should carefully check the documents.

11. This study focused on non-pharmacological labor pain management practice. However, “pharmacological labor pain management practice” was often mentioned in the manuscript. This kind of expression may confuse readers.

7. PLOS authors have the option to publish the peer review history of their article (what does this mean?). If published, this will include your full peer review and any attached files.

Reviewer #3: No

---

## [Author Response · Author response to Decision Letter 6]

14 Sep 2021

First of all, the authors would like to thank Plos one Journal editor and the respective reviewers for reviewing our manuscript and providing the necessary comments to be corrected. As per the comments given, we have made corrections point by point to comment. The authors tried to answer all the issues raised by the editorial team and reviewers. Thank you again for providing this interesting and constrictive comment

---

## [Decision Letter · Decision Letter 7]

10 Feb 2022

PONE-D-20-00957R7Non-Pharmacological Labor Pain Management Practice and Associated Factors among Skilled Attendants working in public health facilities in Gamo and Gofa zone, Southern Ethiopia: A cross-sectional studyPLOS ONE

Dear Dr. Alemu,

Thank you for submitting your manuscript to PLOS ONE. After careful consideration, we feel that it has merit but does not fully meet PLOS ONE’s publication criteria as it currently stands. Therefore, we invite you to submit a revised version of the manuscript that addresses the points raised during the review process. Make the edits in the article according to the review suggestions and have it reviewed by a native English speaker. Then send it to the journal.

We look forward to receiving your revised manuscript.

Kind regards,

Nülüfer Erbil, Ph.D, Prof.

Academic Editor

PLOS ONE

Journal Requirements:

Additional Editor Comments (if provided):

Dear Biresaw Wassihun Alemu,

Make the edits in the article according to the review suggestions and have it reviewed by a native English speaker.

Then send it to the journal.

Reviewers' comments:

Reviewer's Responses to Questions

**Comments to the Author**

1. If the authors have adequately addressed your comments raised in a previous round of review and you feel that this manuscript is now acceptable for publication, you may indicate that here to bypass the “Comments to the Author” section, enter your conflict of interest statement in the “Confidential to Editor” section, and submit your "Accept" recommendation.

Reviewer #3: All comments have been addressed

2. Is the manuscript technically sound, and do the data support the conclusions?

Reviewer #3: Yes

3. Has the statistical analysis been performed appropriately and rigorously? 

Reviewer #3: Yes

4. Have the authors made all data underlying the findings in their manuscript fully available?

Reviewer #3: Yes

5. Is the manuscript presented in an intelligible fashion and written in standard English?

Reviewer #3: No

6. Review Comments to the Author

Reviewer #3: The manuscript has been improved much, but it still needs further revisions by native English speaker.

1.Abstract, Background.It seems better to remove the first sentence from this paragraph and delete “Therefore” in the third sentence.

2.Abstract, Methods. The authors should state directly that this study is a KAP survey. Please make clear the relationship between institutional-based study and sampling technique, and rephrase this paragraph.

3.Abstract, Conclusion. The first sentence, please clarify “.....the practice ......was poor in public health facilities in Gamo and Gofa zone”.

4. There are some inappropriate uses of punctuation marks or grammars in the manuscript. Please carefully check the manuscript. Here are some examples.

(1)Introduction, second paragraph, “Among the practice competencies stated;” , the semicolon could be a comma.

(2)Introduction, second paragraph, “However, its actual practice is not recognized. (15,16)”, delete the symbol “.” .

(3) Methods, “(272) sampled obstetric caregivers;”, “All obstetric caregivers;”, semicolons should be replaced with commas.

(4)Measurement, “Knowledge about Labor pain management methods was measured by a 14-item. Knowledge questionnaire adapted from previous studies”, It should be rephrased as “Knowledge about labor pain management methods was measured by a 14-item questionnaire adapted from previous studies”.

(5)Measurement, “labor pain management practice was measured by 9- checklist items adapted from previous studies”, It should be rephrased as “Labor pain management practice was measured by 9-item checklist adapted from previous studies”.

(6) Discussion, the authors reused “the findings of this study” many times, it is better to use other expressions.There might be other grammar errors in the manuscript, the authors should carefully check the documents.

5. Table 4. Regularly, the reference level should be placed on the first line within a specified subgroup.

6. Discussion. In this section, please do not repeat too much the data in the Results section, just focus on the consistent or inconsistent findings between various investigations, the relationship between knowledge, attitude and practices, and how to improve the practices in the defined areas.

7. PLOS authors have the option to publish the peer review history of their article (what does this mean?). If published, this will include your full peer review and any attached files.

Reviewer #3: No

---

## [Author Response · Author response to Decision Letter 7]

19 Feb 2022

Accompanying this letter, you will find our research article manuscript entitled “Non-pharmacological Labor Pain Management Practice and Associated Factors among Skilled Attendants working in public health facilities in Gamo and Gofa zone, Southern Ethiopia. Institution based cross-sectional study “submitted to your prestigious journal. The research article provides information on Non-pharmacological Labor Pain Management Practice and Associated Factors among Skilled Attendants working in public health facilities in Gamo and Gofa zone. Southern Ethiopia. This article is not yet submitted for any other journals; thus, it is new manuscript to be submitted to this journal for publication. All of the comments were corrected accordingly. All authors have contributed to this manuscript, reviewed and approved the current form of the manuscript to be submitted

We hope the manuscript will suit for publication on this prestigious journal

---

## [Editor Report · Decision Letter 8]

21 Mar 2022

Non-Pharmacological Labor Pain Management Practice and Associated Factors among Skilled Attendants working in public health facilities in Gamo and Gofa zone, Southern Ethiopia: A cross-sectional study

PONE-D-20-00957R8

Dear Dr. Alemu,

We’re pleased to inform you that your manuscript has been judged scientifically suitable for publication and will be formally accepted for publication once it meets all outstanding technical requirements.

Kind regards,

Nülüfer Erbil, Ph.D, Prof.

Academic Editor

PLOS ONE
---

## [Editor Report · Acceptance letter]

13 Apr 2022

PONE-D-20-00957R8 

Non-Pharmacological Labor Pain Management Practice and Associated Factors among Skilled Attendants working in public health facilities in Gamo and Gofa zone, Southern Ethiopia: A cross-sectional study 

Dear Dr. Alemu:

I'm pleased to inform you that your manuscript has been deemed suitable for publication in PLOS ONE. Congratulations! Your manuscript is now with our production department. 

Kind regards, 

on behalf of

Dr. Nülüfer Erbil 

Academic Editor

PLOS ONE